DNA barcode reference library construction and genetic diversity and structure analysis of Amomum villosum Lour. (Zingiberaceae) populations in Guangdong Province

Gong Lu 1 2 3
Zhang Danchun 1 2
Ding Xiaoxia 1
Huang Juan 1 2 3
Guan Wan 1
Qiu Xiaohui qiuxiaohui@gzucm.edu.cn 1 2 3
Huang Zhihai zhhuang7308@163.com 1 2 3
1 The Second Clinical College of Guangzhou University of Chinese Medicine , Guangzhou , China
2 Key Laboratory of Quality Evaluation of Chinese Medicine of the Guangdong Provincial Medicial Products Administration , Guangzhou , China
3 Guangzhou Key Laboratory of Chirality Research on Active Components of Traditional Chinese Medicine , Guangzhou , China
Thiv Mike
Electronic publication date: 2021 Oct 20
Publication date: 2021
Volume: 9
Electronic Location ID: e12325
Received 2021 Jun 8; Accepted 2021 Sep 27
Copyright: ©2021 Gong et al.
Copyright year: 2021
Copyright holder: Gong et al.
License: This is an open access article distributed under the terms of the Creative Commons Attribution License, which permits unrestricted use, distribution, reproduction and adaptation in any medium and for any purpose provided that it is properly attributed. For attribution, the original author(s), title, publication source (PeerJ) and either DOI or URL of the article must be cited.
License URL: https://creativecommons.org/licenses/by/4.0/

Keywords: Amomum villosum, DNA barcode, ISSR, Genetic diversity, Genetic structure

Funding: Guangdong Forestry Department 2017KT1835 Guangdong Provincial Medicial Products Administration 2020ZDB25 002009 Guangdong Provincial Hospital of traditional Chinese Medicine Special Fund YN2019QJ14 The special foundation of Guangzhou Key laboratory 202002010004 This work was supported by grants from Guangdong Forestry Department (2017KT1835), Guangdong Provincial Medicial Products Administration (2020ZDB25, 002009), add a comma Guangdong Provincial Hospital of traditional Chinese Medicine Special Fund (YN2019QJ14) and the special foundation of Guangzhou Key laboratory (202002010004). The funders had no role in study design, data collection and analysis, decision to publish, or preparation of the manuscript.

==============================
Background

Amomum villosum Lour. is the plant that produces the famous traditional Chinese medicine Amomi Fructus. Frequent habitat destruction seriously threatens A. villosum germplasm resources. Genetic diversity is very important to the optimization of germplasm resources and population protection, but the range of inherited traits within A. villosum is unclear. In this study, we analyzed the genetic diversity and genetic structures of A. villosum populations in Guangdong and constructed a local reference DNA barcode library as a resource for conservation efforts.

Methods

DNA barcoding and Inter-Simple Sequence Repeat (ISSR) markers were used to investigate the population genetics of A. villosum. Five universal DNA barcodes were amplified and used in the construction of a DNA barcode reference library. Parameters including percentage of polymorphic sites (PPB), number of alleles (Na), effective number of alleles (Ne), Nei’s gene diversity index (H), and Shannon’s polymorphism information index (I) were calculated for the assessment of genetic diversity. Genetic structure was revealed by measuring Nei’s gene differentiation coefficient (Gst), total population genetic diversity (Ht), intra-group genetic diversity (Hs), and gene flow (Nm). Analysis of molecular variance (AMOVA), Mantel tests, unweighted pair-group method with arithmetic mean (UPGMA) dendrogram, and principal co-ordinates (PCoA) analysis were used to elucidate the genetic differentiation and relationship among populations.

Results

A total of 531 sequences were obtained from the five DNA barcodes with no variable sites from any of the barcode sequences. A total of 66 ISSR bands were generated from A. villosum populations using the selected six ISSR primers; 56 bands, 84.85% for all the seven A. villosum populations were polymorphic. The A. villosum populations showed high genetic diversity (H = 0.3281, I = 0.4895), whereas the gene flow was weak (Nm = 0.6143). Gst (0.4487) and AMOVA analysis indicated that there is obvious genetic differentiation amongA. villosum populations and more genetic variations existed within each population. The genetic relationship of each population was relatively close as the genetic distances were between 0.0844 and 0.3347.

Introduction

Amomum villosum Lour. is a medicinal plant from Zingiberaceae family mainly grown in southern China that produces the famous traditional Chinese medicine (TCM), Amomi Fructus. The dried capsule of A. villosum is brown, ellipsoid, with branched or simple, soft spines that has the effect of dampening appetite, “warming” the spleen to stop diarrhea, regulating qi, and preventing miscarriage (Chinese Pharmacopoeia Commission, 2020). Modern pharmacological studies show that Amomi Fructus has effective anti-ulceration, anti-diarrhea, anti-inflammatory, and antimicrobial properties (Zhu et al., 2018). In addition, Amomi Fructus is also widely used in food, liquors, and tea as a health product and condiment. Volatile oil is regarded as the main active ingredient of Amomi Fructus and it is the quality control component in Chinese pharmacopoeia (Chen et al., 2020). Yangchun City in Guangdong Province is considered the Daodi (genuine) production area of Amomi Fructus due to the high quality of Amomi Fructus from this region. The rapid development of the city and the traditional Chinese medicine business has frequently destroyed the habitat of A. villosum and seriously threatened its germplasm resources. In 2016, Amomi Fructus from Yangchun was designated as one of eight legally protected TCM species in Guangdong Province.

The genetic diversity of a species is the basis for its survival and evolution, which is significant for the analysis of evolutionary polymorphism, genetic relationship, optimization of germplasm resources, and protection of populations. Polymerase Chain Reaction (PCR)-based molecular markers are widely used in the analysis of plant genetic diversity. Among them, Inter-Simple Sequence Repeat (ISSR) markers are fast and efficient, do not require pre-determination of target sequence information, and have the characteristics of high polymorphism, high reliability, and low cost (Grover & Sharma, 2016; Wu et al., 2018). DNA barcodes, a marker proposed in 2003, can be used for both biological identification and genetic diversity analysis (Hebert et al., 2003; Tamboli et al., 2016).

Mitochondrial cytochrome oxidase I (COI) gene is an efficient species identification tool frequently used in the genetic diversity analysis of animals (Manel et al., 2020). In plants, however, low substitution rates of mitochondrial DNA have made it unsuitable, and other barcoding regions were researched as alternatives (CBOL Plant Working Group, 2009). In this study, ISSR and five DNA barcodes including ITS2, psbA-trnH, ITS, matK, and rbcL, were used to investigate the genetic diversity of seven populations of A. villosum in Guangdong Province, especially in Yangchun City, and provide insights into the identification, conservation, breeding, and cultivation of A. villosum.

Material and Methods

Plant material sampling

Sample collections were conducted with the permission of Research Department of the Second Clinical College, Guangzhou University of Chinese Medicine from August 2018 to November 2018. A total of 141 samples from seven A. villosum populations were collected in Guangdong Province using random strategy (Jin & Lu, 2003) that sampling in the largest possible range according to the population size. Six of them were collected in Yangchun City and one was collected in Maoming City. Population ZY was collected from a deserted germplasm garden containing A. villosum plants coming from Guangdong (Jinhuakeng), Yunnan, and Guangxi provinces and the country Myanmar. The sampled plants were identified by Zhihai Huang, the chief Chinese pharmacist of the Second Clinical College of Guangzhou University of Chinese Medicine. Fresh and healthy leaves were removed from the plants, dried, and preserved in silica gel immediately upon collection in the field, then stored in an ultra-low temperature refrigerator (Eppendorf, Hamburg, Germany) in the laboratory. For detailed information and the geographic locations of the samples collected see Table 1 and Fig. 1.

DNA extraction, PCR amplification and sequencing

The total DNA was extracted using a DP305 plant DNA kit (TIANGEN Biotech Co., Ltd., Beijing, China). The NanoDrop2000 ultra-micro ultraviolet spectrophotometer (Thermo Scientific, MIT, USA) was used to determine the DNA concentration and purity. The extracted DNA was finally diluted to 10 ng/L as the template that be subsequently amplified. The PCR amplification reaction system of the experiment contained 2× Taq PCR Mix 12.5 L, forward primer (2.5 M) 1.0 L, reverse primer (2.5 M) 1.0 L, genomic DNA 2.0 L, and added up to 25 L with ddH2O. The primer sequences and amplification conditions of different DNA barcodes are shown in Table S1. All amplification reactions were completed on the ProFlex PCR instrument (Life Technologies, New York, USA). PCR products were sent to Shanghai Majorbio Pharmaceutical Biotechnology Co., Ltd Guangzhou Branch to be sequenced.

ISSR-PCR amplification system

100 ISSR universal primer sequences published by Columbia University were screened (Zhang & Luo, 2004). Six primers that produced clear and reproducible banding patterns were selected (Table 2). ISSR-PCR amplifications were performed on 38 individuals randomly chosen from the seven A. villosum populations (Table 3). The 20 μL ISSR reaction volume included 10 μL2X PCR Mix (containing dye, MgCl2, dNTPs), 2 μL template DNA (20 ng), 1 μLISSR selected primer, and 7 μL ddH2O. The ISSR PCR amplification was programmed in the ProFlex thermocycler as follows: pre-denaturation at 94 °C for 5 min, 35 cycles of denaturation at 94 °C for 45s, annealing at the 46.83–56.60 °C according to the primers for 45s, extension at 72 °C for 2 min, with a final extension at 72 °C for 5 min and preservation at 4 °C. ISSR-PCR products were separated on a 2% agarose gel stained with Goldview by electrophoresis in 1 ×TAE buffer at 80 V. The gels were visualized under UV light and photographed with a ChemiDoc imaging system (Bio-Rad, Hercules, CA, USA). The molecular weights of ISSR-PCR products were estimated using a 100 bp Plus DNA Ladder (TIANGEN Biotech Co., Ltd., Beijing, China).

Table 1 Sampling information of A. villosum populations in Guangdong Province.

No.	Pop.	Serial No.	Longitude /N	Latitude /E	Samples size	
1	ZhongjiaoDong	ZJD	112°04′01″	22°24′16″	33	
2	Tankui Village	TK	112°04′30″	22°24′22″	16	
3	G325 National Roadside	ZY	112°03′59″	22°24′26″	25	
4	XingfuVillage	XFC	112°00′41″	22°21′54″	18	
5	National Geopark	GY	111°49′51″	22°35′19″	19	
6	Dianbai District, Maoming	MM	111°12′21″	21°45′52″	10	
7	Datong Village	YC	111°58′48″	22°23′25″	20	

Figure 1 Geographical distribution of collected A. villosum populations.

Populations in the small red box are enlarged in the large red box.

Table 4 Efficiency of PCR and sequence characterization of DNA barcodes.

	Numbers of sequences	PCR amplification success rate (%)	Sequencing success rate (%)	Average sequence length (bp)	GC content (%)	Conservative sites	
ITS2	134	95.74	95.04	229	59.83	229	
psbA-trnH	76	57.45	53.90	667	28.49	667	
ITS	139	99.29	98.58	653	56.20	653	
matK	41	29.08	29.08	787	29.22	787	
rbcL	141	100.00	100.00	729	41.84	729	

Table 2 ISSR banding patterns of seven A. villosum populations.

Primers	Sequences	Tm(°C)	Amplification bands	Polymorphic bands	PPB(%)	
UBC808	(AG)8C	52.00	11	11	100.00	
UBC817	(CA)8A	49.24	11	10	90.91	
UBC825	(AC)8T	52.00	12	12	100.00	
UBC840	(GA)8YT	52.74	12	10	83.33	
UBC866	(CTC)6	56.60	6	4	66.67	
UBC889	DBD-(AC)7	46.83	14	9	64.29	
sum	—	—	66	56	84.85	
mean	—	—	11	9.33	84.82	

Data analysis

The two-way sequenced peaks of DNA barcodes were evaluated and assembled using the CondonCode Aligner v8.0.1 software (Bonfield, Cristina & Rodger, 1998). Low-quality areas at both ends of the assembled sequences were removed. ITS2 barcodes were annotated by cutting off the conserved 5.8S and 28S motifs based on HMM (Keller et al., 2009) at the ITS2 database (Ankenbrand et al., 2015). Mega6.0 software was used to align DNA barcode sequences and calculate sequence statistics including the base composition ratio, GC content, heterotopic site information, conservative site, and parsimony informative sites (Tamura et al., 2013). Haplotype sequences for each barcode were exhibited in the quick response (QR) code picture. In the picture, each vertical line represented a nucleotide base and the QR code on the right could be scanned directly for the DNA sequence.

Reproducible ISSR-PCR bands were determined with the help of the GelPro32 software and manual correction. These clear bands were scored as either present (1) or absent (0), thus generating an ISSR phenotype data matrix that was imported in Popgene32 software to analyze genetic diversity and genetic structure (Yeh et al., 1999). Genetic diversity parameters including percentage of polymorphic sites (PPB), number of alleles (Na), effective number of alleles (Ne), Nei’s gene diversity index (H), and Shannon’s polymorphism information index (I) were calculated. Genetic structure parameters including Nei’s gene differentiation coefficient (Gst), total population genetic diversity (Ht), intra-group genetic diversity (Hs), and gene flow (Nm) were calculated. GenAlEx 6.502 software was used to estimate the components of genetic variance within and among populations by analysis of molecular variance (AMOVA) and to assess the correlation between population genetic distance and geographic distance by Mantel tests (Guillot & Rousset, 2013). Genetic distances among populations were calculated and a UPGMA dendrogram was constructed by using NTSYS 2.10e (Rohlf, 2000).

Results and analysis

DNA barcode reference library construction

We extracted the genomic DNA from 141 samples of A. villosum. The OD260/280 was 1.76−1.98 for all the DNA samples and the concentration was 73.70–1294.80 ng/L. Five DNA barcodes of all the samples were amplified and sequenced bi-directionally. Finally, 531 sequences were gained and submitted to the Genbank database (Table S2). The PCR amplification and sequencing results showed that the success rate of sequencing for each barcode was rbcL (100.00%) > ITS (98.58%) > ITS2 (95.04%) > psbA-trnH (53.90%) >matK (29.08%) (Table 4). The ranking of the success rate of PCR amplification was consistent with that of sequencing. Thus, these sequences constructed a DNA barcode reference library for A. villosum in Guangdong Province.

Table 3 Genetic diversity of A. villosum populations based on ISSR.

	NO.	Na	Ne	H	I	PPB(%)	
ZJD	9	1.5303	1.3492	0.1998	0.2954	53.03	
TK	4	1.2879	1.1848	0.1117	0.1658	28.79	
ZY	7	1.7121	1.4240	0.2536	0.3816	71.21	
XFC	5	1.4394	1.2685	0.1624	0.2434	43.94	
GY	5	1.4848	1.3516	0.1998	0.2897	48.48	
MM	3	1.4901	1.3273	0.1818	0.2604	40.91	
YC	5	1.4394	1.2753	0.1648	0.2460	43.94	
mean	7	1.4834	1.3115	0.1820	0.2689	47.19	
At species level	38	1.9394	1.5662	0.3281	0.4895		
Notes.

Na number of alleles

Ne effective number of alleles

H Nei’s gene diversity index

I Shannon’s polymorphism information index

PPB percentage of polymorphic sites

We also analyzed the sequences of each barcode we obtained. The sequences of the five barcodes had no variation sites, showing a strong conservation. Among the five DNA barcodes, ITS2 had the shortest sequence length and the highest GC content. The sequences of each barcode are shown in Fig. 2.

Figure 2 DNA barcode sequences of A. villosum.

(A) ITS2, (B) psbA-trnH, (C) ITS, (D) matK, (E) rbcL.

ISSR polymorphism and genetic diversity

We established the ISSR-PCR system for A. villosum. The ISSR-PCR fragments of A. villosum populations ranged from 300 bp to 2000 bp. An example of amplified profiles is shown in Fig. S1. All the ISSR-PCR profiles were deposited in Figshare (DOI: 10.6084/m9.figshare.14622987). A total of 66 ISSR bands were generated from A. villosum populations using the selected six primers. Among them, 56 bands were polymorphic, thus the percentage of polymorphic bands was 84.85% for all seven A. villosum populations. For each primer, it amplified 6–14 bands with the polymorphic ratio of 64.29%–100.0% (Table 2).

Genetic diversity information of A. villosum populations analyzed by Popgene 32 were shown in Table 3. Among the seven populations of A. villosum, population ZY had the highest genetic diversity (PPB =71.21%, Na =1.7121, Ne =1.4240, H = 0.2536, I = 0.3816) while population TK had the lowest genetic diversity (PPB =28.79%, Na =1.2879, Ne =1.1848, H = 0.1117, I = 0.1658). At the species level, the values of Na, Ne, H, and I were 1.9394, 1.5662, 0.3281, and 0.4895, respectively.

Genetic structure of A. villosum populations

Genetic structure information of A. villosum populations analyzed by Popgene 32 is shown in Table 5. Total genetic diversity (Ht) of the seven populations was 0.3299, while within the population genetic diversity was 0.1819. The gene differentiation coefficient (Gst) was 0.4487, indicating that 55.13% of the genetic variation existed within populations. This result was similar to the molecular variance analysis (AMOVA), which showed 68.74% (P = 0.001) genetic variation was within populations while 31.26% (P = 0.001) was between populations (Table 6). Additionally, the gene flow (Nm) among different populations was 0.6143.

Table 5 Genetic structure of A. villosum populations based on ISSR.

Samples size	Genetic diversity of the total populations (Ht)	Genetic diversity within populations (Hs)	Gene differentiation coefficient (Gst)	Gene flow (Nm)	
38	0.3299	0.1819	0.4487	0.6143	
Standard deviation	±0.0272	±0.0102	—	—	

Table 6 AMOVA analysis of A. villosum populations based on ISSR.

Source of variation	Degree of freedom /df	Mean square deviation /SS	Mean square value /MS	Variance component	Variance component percentage (%)	P	
Among populations	6	167.835	27.972	3.718	31.26	0.001	
Within populations	37	253.455	8.176	8.176	68.74	0.001	
Sum	43	421.289	—	11.894	100	—	

Genetic distance and cluster analysis

Genetic distance is the main indicator used to examine the degree of genetic differentiation and the relationship between groups. The genetic distances of the seven populations were between 0.0844 and 0.3347 (Table 7). Among them, the smallest genetic distance was between the ZJD and the TK populations (0.0844), and the largest one was between the XFC and the YC populations (0.3347). Mantel tests carried out with NTSYS-pc 2.0 indicated that genetic distance and geographical distance were not significantly correlated (r = 0.02698, P = 0.5504) (Fig. 3).

The UPGMA clustering map of populations based on the genetic similarity coefficient was constructed using Nei’s genetic distances of the seven populations (Fig. 4). The A. villosum populations were divided into three groups at the similarity coefficient of 0.84. Three populations, ZJD, TK, and ZY, formed one group. Three populations, GY, MM, and YC, formed another group. One population, XFC, formed a single group. The results of PCoA based on the unbiased matrix of Nei were consistent with UPGMA cluster analysis. The first two principal components accounted for 48.04% of the total variation of ISSR markers indicated from PCoA (Fig. 5).

Table 7 Nei’s genetic distance of A. villosum populations.

	ZJD	TK	ZY	XFC	GY	MM	YC	
ZJD	0.0000							
TK	0.0844	0.0000						
ZY	0.0973	0.1247	0.0000					
XFC	0.2347	0.3317	0.1957	0.0000				
GY	0.2260	0.2697	0.1946	0.1463	0.0000			
MM	0.2785	0.2827	0.2218	0.2342	0.1132	0.0000		
YC	0.2697	0.2719	0.2176	0.3347	0.2003	0.1382	0.0000	

Figure 3 Correlation of geographic distance and genetic distance.

Figure 4 UPGMA clustering map of A. villosum populaitons.

Figure 5 PCoA map of A. villosum populations (coord. 1 = 48.04% and coord. 2 = 26.74%).

Discussion

Many DNA barcode reference libraries have been constructed for the purpose of more rapid and accurate species identification (Gong et al., 2018; Mosa et al., 2018). In this study, five DNA barcodes, ITS2, psbA-trnH, ITS, matK and rbcL, were amplified and sequenced from 141 individuals from seven A. villosum populations and resulted in the identification of 531 sequences. Thus, the first local DNA barcode reference library of A. villosum in Guangdong Province was constructed. Fruits from the same genus plants, especially A. xanthioides Wall. ex Baker and A. longiligulare T.L. Wu, are very morphologically similar to A. villosum’ s and are also genuine sources of Amomi Fructus. In addition, other adulterants of Amomi Fructus are sold in the markets (Ch et al., 2018). Huang et al. (2014) used seven DNA barcodes to identify the three genuine germplasms of Amomi Fructus. An ideal DNA barcode should be easily retrievable, bi-directionally sequenced, and provide maximal discrimination among species (CBOL Plant Working Group, 2009). In this study, the rbcL gene had the highest PCR amplification and sequencing success rate of the five A. villosum DNA barcodes. For a more comprehensive assessment of the discrimination power of DNA barcodes, adulterants of A. villosum are needed to asses together.

DNA markers, especially ITS sequences, are used in plant population genetic analysis (Li et al., 2020). In this study, we aligned the sequences within the DNA barcodes, but no variable sites of each DNA barcode were found. Therefore, genetic diversity could not be analyzed by these DNA barcode markers. This might have been caused by the narrow geographic collection of A. villosum populations (Phillips, Gillis & Hanner, 2019). Though, intra-specific divergence can occur at a very high rate within geographically isolated populations (Hebert & Ryan, 2005), it seems that the geographic areas need to be wide for the populations to be analyzed for genetic diversity by DNA barcodes (Poyraz, 2016). Species identification and genetic diversity are two conflicting problems for the barcoding approach and these problems have to be addressed during the library construction stage (Sarwat & Yamdagni, 2014). Due to limited sampling, how much variation is actually needed to separate species is unknown (Phillips, Gillis & Hanner, 2019) and requires further research.

We analyzed the genetic diversity and genetic structure of A. villosum populations using ISSR markers through the whole genome. The results indicated that germplasm materials of A. villosum in Yangchun had relatively higher genetic diversity (H = 0.3281, I = 0.3816). Among the seven populations, population ZY had the highest genetic diversity; this is consistent with the fact that it had diverse germplasm resources. The Gst results (0.4487) indicated that there is obvious genetic differentiation among A. villosum populations (Wright, 1984) and more genetic variations exist within populations. This was also confirmed by the AMOVA analysis. The gene flow value (Nm =0.6143) revealed that genetic drift was the main cause for genetic variations among populations (Slatkin, 1987). The weak gene flow of A. villosum populations is probably due to its pollination style and habitat fragmentation. The morphological structure of A. villosum flowers makes them extremely difficult to self-pollinate. Due to the lack of pollinating insects, such as bees and ants A. villosum in Guangdong mainly relies on artificial pollination. The gene flow is consequently restricted to a small population. The habitat fragmentation of A. villosum in Yangchun is also likely to reduce the exchange of pollens between populations. Xu & Ding (2005) used RAPD markers to assess the three populations of A. villosum including two populations from Yangchun city, Guangdong province and one population from Yunnan province. They found the high genetic diversity of A. villosum populations with the polymorphic percentage was 92.16%, which was consistent with our finding.

Amomi Fructus has been used in China for more than a thousand years and was mainly imported from abroad for much of that time. A. villosum has been cultivated in Yangchun for approximately 200 years after being introduced into Guangxi, Yunnan, and Fujian provinces in southern China (Caiying, Ruoting & Xiaoping, 2011). Species with high genetic variation can resist survival pressures caused by various environmental changes. Amomum villosum populations in Guangdong Province, especially in the Daodi producing area of Amomi Fructus, Yangchun City, have high genetic diversity inferred from our study. Yangchun has expanded the planting area of Amomum villosum in recent years. However, considering the small populations and the increasingly fragmented habitat as well as the lower gene flow of A. villosum, we need to take measures to protect the genetic resources of the species. On one side, we could conduct in-situ conservation in Tankui Village, because the population has the lowest genetic diversity indicating it was the most threatened. On the other side, we recommend collecting core germplasm resources based on this study and reconstructing the germplasm garden in Yangchun City to preserve genetic diversity of A. villosum.

Conclusion

In this study, a local DNA barcode reference library containing 531 sequences was constructed from 141 samples from seven A. villosum populations in Guangdong Province. The A. villosum populations have high genetic diversity, but ISSR markers revealed that their gene flow is weak. The genetic relationship of each population is relatively close. More measures are needed to protect the genetic resources of A. villosum in Guangdong province, especially in Amomi Fructus’s Daodi producing area, Yangchun City.

Supplemental Information

Supplemental Information 1 ISSR-PCR electrophoretic maps of A. villosum populations with primer UBC808

Click here for additional data file.

Supplemental Information 2 The primers and PCR conditions of five DNA barcodes

Click here for additional data file.

Supplemental Information 3 Genbank accession numbers of DNA barcodes

Click here for additional data file.

The authors would like to thank Runfa Li, the technical director of TamKan Agricultural development Co., Ltd for the help with sampling. We also thank Baosheng Liao for providing the program that generated the pictures of the DNA barcodes.

Additional Information and Declarations

Competing Interests

Author Contributions

Field Study Permissions

DNA Deposition

Data Availability

The authors declare there are no competing interests.

Lu Gong, Xiaohui Qiu and Zhihai Huang conceived and designed the experiments, authored or reviewed drafts of the paper, and approved the final draft.

Danchun Zhang performed the experiments, analyzed the data, prepared figures and/or tables, authored or reviewed drafts of the paper, and approved the final draft.

Xiaoxia Ding and Wan Guan performed the experiments, analyzed the data, prepared figures and/or tables, and approved the final draft.

Juan Huang analyzed the data, authored or reviewed drafts of the paper, and approved the final draft.

The following information was supplied relating to field study approvals (i.e., approving body and any reference numbers):

Research Department of The Second Clinical College, Guangzhou University of Chinese Medicine.

The following information was supplied regarding the deposition of DNA sequences:

A total of 531 sequences are available at GenBank (Table S2). The haplotype of these sequences for each barcode are shown in Fig. 2.

The following information was supplied regarding data availability:

All the ISSR-PCR profiles are available in Figshare: Gong, Lu (2021): ISSR-PCR electrophoretic maps of A. villosum populations. figshare. Figure. https://doi.org/10.6084/m9.figshare.14622987.v1.

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
