# Peer review of "DNA barcode reference library construction and genetic diversity and structure analysis of Amomum villosum Lour. (Zingiberaceae) populations in Guangdong Province"

_PeerJ, doi:10.7717/peerj.12325_

## Round 0.1 · original submission · Minor Revisions

Dear authors

I received 2 reviews on your manuscript. Both are generally positive but several things need to be modified.

- More information in material and methods should be added (see reviewers).

- Add information on the phytochemistry and morphology of the taxon

- Compare your outcomes with other published works in the same field!

- Discuss in more detail what your results mean for conservation. Which populations are most threatened? What needs to be done where?

Please refer to all points made by the reviewers. I assign minor revisions to your manuscript.

Best wishes
Mike Thiv

·

Basic reporting

In this study, the authors analyzed the genetic diversity and genetic structures of A. villosum populations and they used DNA barcoding and Inter-Simple Sequence Repeat (ISSR) markers for investigation of the population genetics of A. villosum.

Experimental design

Your experimental needs more detail. I suggest that you improve the description at lines 81- 92 to provide more justification for your study.
-The methods of collection and references is needed.
-For example the voucher number of each population is needed.
-The quality of image is very low and needs to improve.

Validity of the findings

However the genetic diversity is very important in medicinal plant but the phytochemical and morphological characterization is also need. For this reason I highly suggested that this detail must be added to this paper.

Additional comments

Your experimental needs more detail. I suggest that you improve the description at lines 81- 92 to provide more justification for your study.
-The methods of collection and references is needed.
-For example the voucher number of each population is needed.
-The quality of image is very low and needs to improve.
However the genetic diversity is very important in medicinal plant but the phytochemical and morphological characterization is also need. For this reason I highly suggested that this detail must be added to this paper

·

Basic reporting

no comment

Experimental design

no comment

Validity of the findings

no comment

Additional comments

The manuscript entitled "DNA barcode reference library construction and genetic diversity and structure analysis of Amomum villosum Lour. populations in Guangdong Province" reports genetic diversity and populations genetic structures of collected A. villosum from Guangdong. The data, in the complex, contribute to increase the knowledge about the genomic structure and give correct information about the dimension of the gene pool useful to define the core collection to preserve genetic diversity.The manuscript needs to be re-read and there are a few questions and errors that should be answered and corrected by the authors:
Abstract
1. Background: Line 23: Please replace "Ammomum villosum Lour" by "Amomum villosum Lour".
2. Results: Line 43-44: "…56 bands, 84.85% for all the seven A. villosum populations, were amplified polymorphic". In this sentence, it is better to delete the word "amplified".
Materials and Methods
Plant Material Sampling: Line 81-82: "…Research Department of The Second Clinical College". Word "The" in this sentence should be written in non-capital.
Plant Material Sampling: For some populations, ten individuals and for another population 33 individuals have been selected. It is better to explain on what basis the number of samples are different?

ISSR-PCR amplification system Line 113: Please specify the template DNA concentration.
Discussion
In the discussion section, you could refer to other genetic diversity work done on this plant or nearby species.
Figure:
The populations in Figure 1 should be completely separated and be clearly marked on the map.
Table 4: Please explain the abbreviations (Na, Ne, H and …) below the table.

The whole text is not justify and in in somewhere the space between words and lines is not suitable.

---

## Round 0.2 · accepted · Accept

Dear authors

I regard your ms. as acceptable.
I found several smaller points which still should be changed.
Best wishes
Mike Thiv

title: add family in brackets.
l. 57 was => is
l. 59 add reference
l. 109 rewrite
l. 239 Fruits
l. 242 markets
l. 243 Huang add year
l. 245 The rbcL gene ...
l. 252 Though, ...
l. 269 state which insect sare pollinators
l. 272 seeds (no spore plant!!)
l. 272 Xu & Ding add year
l. 274 They found
l. 278 Amomum villosum ...
l. 281 Amomum villosum ...